# Effect of Solid Volume Concentration on Rheological Properties of Chengdu Clay Slurry

**Xianjun Ji** [1,*] **, Ying Liang** [2] **and Wenhao Cao** [1]

1 School of Civil Engineering, Nanyang Institute of Technology, Nanyang 473004, China; caowenhao0325@163.com
2 School of Mathematics and Physics, Nanyang Institute of Technology, Nanyang 473004, China; liangying589@163.com
* Correspondence: jifeng988@163.com; Tel.: +86-15139091660

**Abstract:** The determination of rheological model about the debris flow is the basis of the simulation of mud flow impact distance and sedimentary fan. By using a mcr301 rheometer, rheological experiments of Chengdu clay slurry with different solid volume concentrations were carried out and the effect of solid volume concentration on shear stress were analyzed. Then the rheological process of Chengdu clay slurry with different solid volume concentration was fitted on the basis of the power law model, the Bingham model and the H–B model. The conclusions are drawn as follows: Chengdu clay mud is a typical shear-thinning non-Newtonian body. The influence of solid concentration on the flow curve is different. When the solid volume concentration is not less than 34% and the shear rate is less than $1.0 \text{ s}^{-1}$, the shear stress increases rapidly as the shear rate increases. Meanwhile, when the shear rate is greater than $1.0 \text{ s}^{-1}$ the shear stress decreases with the increase in the shear rate. When the solid volume concentration is less than 31.6% and the shear rate is less than $1.0 \text{ s}^{-1}$, the shear stress increases with the increase in the shear rate, while when the shear rate is more than $1.0 \text{ s}^{-1}$, the shear stress is less affected by shear rate. In the range of low shear rate (less than $1.0 \text{ s}^{-1}$), the increase amplitude of shear stress (slope of semi logarithmic coordinate flow curve) increases as the solid volume content increases. The flow curve of Chengdu clay mud can be reflected in the whole process by using the Herschel and Bulkley model. It is the best mathematical model to fit the rheological process of Chengdu clay mud. According to the above results, the effect of solid volume concentration on the yield stress of the H–B model is analyzed.

**Keywords:** Chengdu clay mud; solid volume concentration; rheological experiment; rheological model

## 1. Introduction

Due to climate change, extreme weather such as rainstorm occurs all over the world, and landslide and debris—flow disasters induced by rainstorm—also occur frequently. After slope instability, a coarse particle fluid mixtures composed of inviscid coarse particles and viscous mud (fine-grained soil water mixture) is formed, which is in a viscous state (viscous debris flow) during its movement process, and its flow is similar to that of fluid. The authors of [1–4] pointed out that for viscous debris flow, coarse particles are dispersed in the mud. During the movement process, coarse particles interact with the mud, and its flow behavior is controlled by the rheological properties of the mud matrix. The mud moves with coarse particles, so the debris flow has strong impact and great destruction [5]. For example, during the "11.02" landslide and debris flow disaster in Chuxiong Prefecture, Xinhua Village was buried within 3 min [6]. Therefore, the cognition of mud rheological properties is very important to understand and debris flow movement, including applicable rheological models, yield stress, plastic viscosity and flow index, etc., should be analyzed. This is the basis of determining the velocity, impact force and outflow distance of debris flow, and it is helpful to evaluate the risk related to debris flow movement.

Yield stress and plastic viscosity are the basis of simulating debris flow movement. Lots of research work has been performed by relevant scholars, and they mainly focused on the influence of solid concentration on yield stress and apparent plastic viscosity. [7] investigated the rheological characteristics of debris flow in the Gargano basin and analyzed the variation process of plastic viscosity and yield stress of mud with different solid volume concentration under different shear rates. The reconstituted samples of fine-grained natural debris flow were investigated and the effects of sediment content, yield strength and plastic viscosity were analyzed [8]. The rheological behavior reconstituted samples composed of silt and clay with variable sand content about the debris flow were investigated [9]. Based on the results of rheological experiments, it was concluded that the size and percentage of coarse particles greatly affect the rheological behavior and rheological parameters of debris flow slurry. The rheological behaviors of spherical and irregular plastic granular materials in glycerol aqueous solution were studied and a general rheological model considering particle morphology was provided [10,11]. The influence of water content on the rheological properties of Chengdu clay slurry was analyzed and then the relationship between the peak value of initial shear and complex shear stress and water content was established [12]. In addition, through rheological experiments, an empirical relationship to estimate the rheological properties of mud according to the liquid index was proposed [13–15].

A proper rheological model of debris flow is the basis of simulating the impact distance of mud flow and sedimentary fan. Due to the differences in debris flow composition all over the world, the applicable rheological model is often discussed in the investigation on the debris flow. Common rheological models include the power law model [16,17], the Bingham model [18–21], and the Herschel–Bulkley model [19,22]. In all, the Bingham model has been widely applied, but there is obvious dispersion among the results of many experiments. The Herschel–Bulkley model is suggested for adoption. Debris flow is a suspended fluid composed of viscous mud and non-viscous coarse particles. In addition to the collision between particles, the movement of viscous debris flow is controlled by mud. So Chengdu clay with fine particle composition is selected as the experimental material, the flow curve test of Chengdu clay slurry with different solid concentrations is tested in this paper, and the experimental results of different models are modeled and fitted to determine the best rheological model of Chengdu clay slurry, providing a reference for the analysis of debris flow movement process and the prevention and control of debris flow disaster in this area.

## 2. Rheological Model

According to the rheological characteristics of mud, the power law model (Equation (1)), Bingham model (Equation (2)) and Herschel and Bulkley model (Equation (3)) are applied to analyze the rheological process of Chengdu clay mud.

$$\tau = \eta \times \dot{\gamma}^{n} \tag{1}$$

$$\tau = \tau_{B} + \eta_{B} \times \dot{\gamma} \tag{2}$$

$$\tau = \tau_{HB} + \eta_{HB} \times \dot{\gamma}^{n} \tag{3}$$

The power law model (Equation (1)) is applicable to non-Newtonian fluids without yield stress. When $n = 1$, it is Newtonian fluid; When $n > 1$, shear becomes diluent; When $n < 1$, the shear becomes thinner. The Bingham model (Equation (2)) is applicable to fluids with yield stress. When the shear stress is less than the yield stress $\tau_{B}$, only elastic deformation occurs; when the shear stress is greater than the yield stress $\tau_{B}$, the shear stress increases linearly with the increase in shear rate, which is a Newton body. The Herschel and Bulkley model (Equation (3)) is applied to non-Newtonian fluids with certain yield stress. When $n > 1$, it is a shear thickening fluid with dilatancy; when $n < 1$, it is shear thinning fluid with pseudoplasticity; when $n = 1$, it is the Bingham model. The above three models (1, 2, 3), $\tau$ is shear stress, $\tau_{B}$, $\eta_{B}$ are Bingham yield stress and viscosity coefficient, respectively,

$\tau_{HB}$, $\eta_{HB}$ are the yield stress and viscosity coefficient of HB model, respectively; $\eta$ is the flow coefficient of the power law model; $n$ is the flow index; $\dot{\gamma}$ is the shear rate.

## 3. Testing Materials and Equipment

### 3.1. Testing Materials

Chengdu clay is widely distributed in the second and third terraces of Minjiang River, the eastern suburb of Longquanyi in Chengdu. It is brownish yellow, brownish yellow and grayish yellow, hard plastic in texture, containing iron and manganese nodules and strong viscosity [23]. Take some clay from Longquan District of Chengdu, which is brownish yellow. According to the experimental method of liquid plastic limit detecting, the liquid plastic limit is determined by using the liquid plastic limit joint tester. The water content of the liquid limit is 66.5% and the water content of the plastic limit is 24.5%. The plasticity index Ip is 42; it belongs to clay, which is consistent with the literature [24]. The composition of Chengdu clay particles is analyzed by a laser diffraction particle size analyzer (Malvin Mastersizer 2000, Worcestershire, UK) (in Figure 1). The content of quartz particles greater than 20 is nearly 30%. The density of Chengdu clay particles determined by using volumetric flask method is ds = 2.7 g/cm$^3$. Chengdu clay particles are sub angular and sub circular, with a sorting coefficient of 1.24 and a skewness of 0.65 [25].

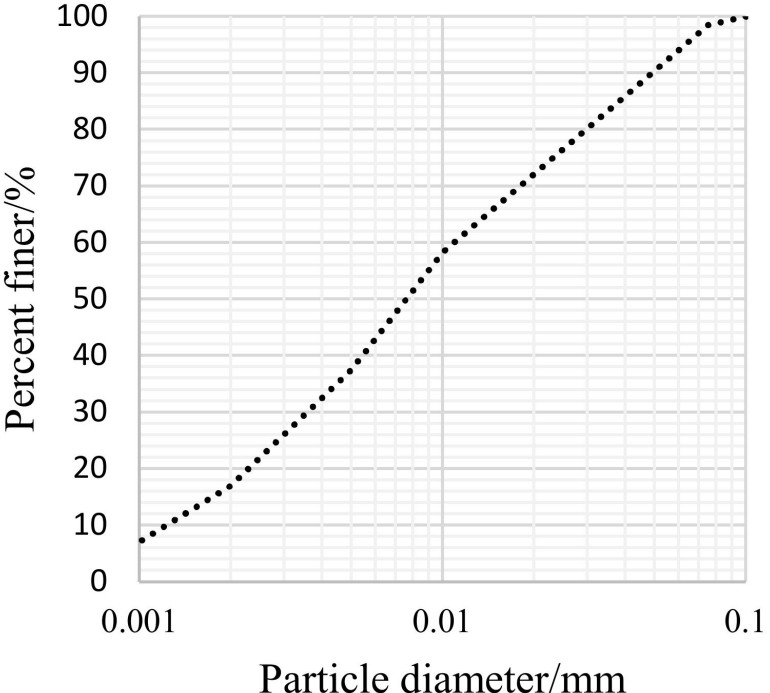

**Figure 1.** Grain size distribution of the Chengdu clay materials.

The soil sample is put into the container, then it is soaked in water for more than 48 h, mixed manually, and fully saturated for standby. The saturated back-up Chengdu clay slurry is taken and its initial water content is determined by using the drying method ($w_0$: mass of water per unit volume and mass percentage of dry soil). Then it is placed in 10 measuring cups, different amounts of distilled water are added to the container, respectively, and the water content $w_1$ is calculated after water is added according to the initial water content $w_0$. The slurry solid volume concentration is the ratio of solid volume ($V_s$) to total volume $V$ (solid volume $V_s$ and water content volume $V_v$: $V = V_s + V_v$) (expressed as a percentage), i.e.,:

$$C_s = \frac{V_s}{V} \times 100\% = \frac{V_s}{V_s + V_v} \times 100\% \qquad (4)$$

According to water content $w_1$ and particle density ds, the solid volume content of mud in each container is calculated in Table 1.

**Table 1.** Solid volume concentration of the Chengdu clay mud.

| Slurry Sample | 1 | 2 | 3 | 4 | 5 | 6 | 7 | 8 | 9 | 10 |
|---|---|---|---|---|---|---|---|---|---|---|
| solid volume concentration/% | 42.6 | 39.4 | 35.6 | 34.0 | 31.6 | 26.8 | 22.0 | 16.9 | 10.9 | 8.5 |

### 3.2. Experimental Equipment

The rheological experiment was measured by MCR301 Rheometer in Antonpa, Austria. The Rheometer adopts air bearing. The torque range is 0.002 μ Nm~200 mNm. The deflection angle (preset) range is 0.1 to ∞ μ rad. The speed range is (CSS) $10^{-7}$~3000 rpm. The frequency range is $10^{-5}$–102 Hz. The normal stress range is 0.01–50 N, and the normal stress accuracy is 0.002 N. The temperature range is −150 °C~1000 °C.

The rotary rheometer used in this experiment is equipped with a blade rotor geometry system. The system is composed of four thin blades. It is arranged equiangularly around a small cylindrical axis. The diameter of blade is 22 mm, the height $L$ of blade is 16 mm, and the inner diameter of the outer cylinder is 42 mm. During the experiment, the rotor of the blade is immersed in the mud cylinder and rotated coaxially. The serious slip effect of the wall can be avoided in the experimental system [26], and the inertial effect and normal stress difference are ignored. The shear stress and shear rate of materials can be calculated by the following formula:

$$\tau = \frac{T}{2\pi R_1^2 L} \tag{5}$$

and

$$\dot{\gamma} = \frac{\Omega R_1}{(R_2 - R_1)} \tag{6}$$

where $T$ is the torque applied to the blade and $\Omega$ is the angular velocity of the blade rotor.

### 3.3. Experimental Method

Based on the observation of debris flow disaster, [27,28] pointed out that the flow velocity of debris flow is usually between 0.5 and 20 m/s, and the average strain rate of debris flow is usually between 0.1 and 20 $s^{-1}$. Therefore, the strain-scanning experiment is mainly carried out in this experiment (obtaining the flow curve by increasing the shear rate). In the experiment, the shear rate (blade rotor rotation rate) is increased logarithmically from 0.01 $s^{-1}$ to a larger (maximum) value (30 $s^{-1}$), and the corresponding shear stress was measured.

Ref. [21] pointed out that the uniform fluid with the behavior of recombinant debris flow can be obtained only in a range of specific solid concentration. Below this range, the material will settle rapidly; above this range, it behaves like a solid. In order to reduce the influence of particle settlement on rheological experiment, the slurry is fully stirred for 10 min before the test is performed to homogenize the slurry. Then the mud is put into the outer barrel of the rheometer and the rotor of blade is put below the mud surface in the barrel. The rotation rate of the blade rotor is set according to the predesign value, and the strain-scanning experiment (at the beginning of the sweeping test, the material in the cylinder is at rest) is performed. The time from mud loading to the beginning of the experiment is controlled within 3 min. According to the research [29], for Chengdu clay mud with high content of fine particles, the influence of particle settlement on rheological experiment can be ignored. In all experiments, the test temperature of the sample was kept constant at 20 °C in the water circulation system.

## 4. Results and Discussion

### 4.1. Effect of Solid Volume Concentration on the Flow Curve of Mud

In this paper, rheological experiments of Chengdu clay slurry with different volume concentrations are carried out, and the change process of shear stress and shear rate of Chengdu clay slurry is plotted in a semi-logarithmic coordinate system (in Figure 2a–d). The solid volume concentrations were 42.0%, 39.4% (Figure 2a), 35.6%, 34% (Figure 2b), 31.6%, 26.8%, 22.0% (Figure 2c), 16.9%, 10.9% and 8.5% (Figure 2d), respectively. It can be found that all Chengdu clay slurries show non-Newtonian behavior, and the shear stress level increases with the increase in slurry solid concentration. However, the influence of mud concentration on the flow curve is quite different: (1) for Chengdu clay mud with a solid volume concentration no less than 34%, when the shear rate is less than $1.0/s^{-1}$, the shear stress increases rapidly with the increase in shear rate, and the increase in shear stress (slope of semi logarithmic coordinate flow curve in the range of low shear rate) increases with the increase in solid volume content; when the magnitude of shear rate is more than $1.0\,s^{-1}$, the shear stress decreases with the increase in shear rate. In the semi-logarithmic coordinate systems of shear stress and shear rate, the upward-bending radian of the flow curve increases with the increase in solid volume concentration. (2) For Chengdu clay slurry with solid volume concentration less than 31.6%, when the magnitude of shear rate is less than $1.0\,s^{-1}$, the shear stress increases with the increase in shear rate, but the increasing range of shear stress (slope of semi logarithmic coordinate flow curve in the range of low shear rate) decreases with the decrease in solid volume content; when the magnitude of shear rate is more than $1.0\,s^{-1}$, the shear rate has little effect on the shear stress. According to the research results of Bentonite and Illite natural clay, suggested by [30], its flow behavior depends not only on the shear rate, but also on the particle concentration. The whole process shows shear-thinning characteristics [31,32].

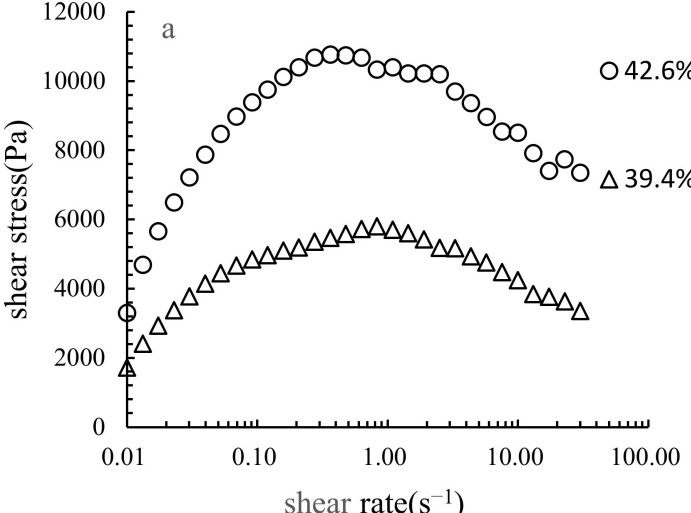

**Figure 2.** *Cont*.

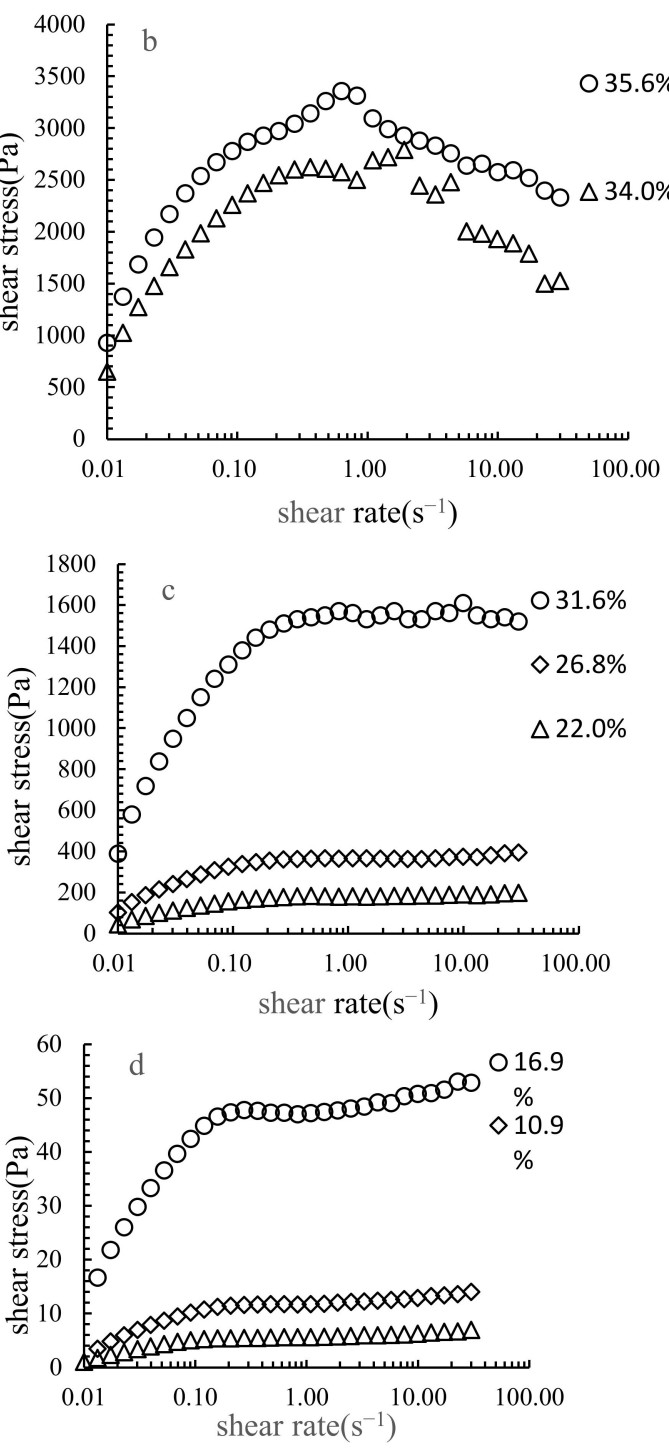

**Figure 2.** The shear stress vs. shear rate. The solid volume concentrations were 42.0%, 39.4% (**a**), 35.6%, 34% (**b**), 31.6%, 26.8%, 22.0% (**c**), 16.9%, 10.9% and 8.5% (**d**), respectively.

*4.2. Rheological Model of Chengdu Clay Slurry*

The rheological model is the basis of simulating mud and debris flow movement. In fact, for heterogeneous mixtures, such as mud, there is interaction between solid particles and surrounding fluid, but it can be regarded to be similar to the microstructure behavior of homogeneous non-Newtonian fluid; shear stress is a function of shear rate. According to the rheological experimental results of Chengdu clay mud, the results demonstrate that different rheological behaviors are displayed in different shear rate ranges: for mud with solid volume concentration greater than 34%, if the magnitude of shear rate is less

than 1 s$^{-1}$ (low shear rate), the shear stress increases as the increase in shear rate; if the magnitude of shear rate is more than 1 s$^{-1}$ (high shear rate), the shear stress decreases with the increase in shear rate. It is difficult to describe its rheological process using a specific model. However, the change process of shear stress at low shear rate can be applied to represent the start-up process of mud flow and debris flow, which is very important for the start-up and early warning of debris flow. Therefore, different models are applied to fit the low shear rate (shear rate is less than 2 s$^{-1}$ rheological process) (Figure 3).

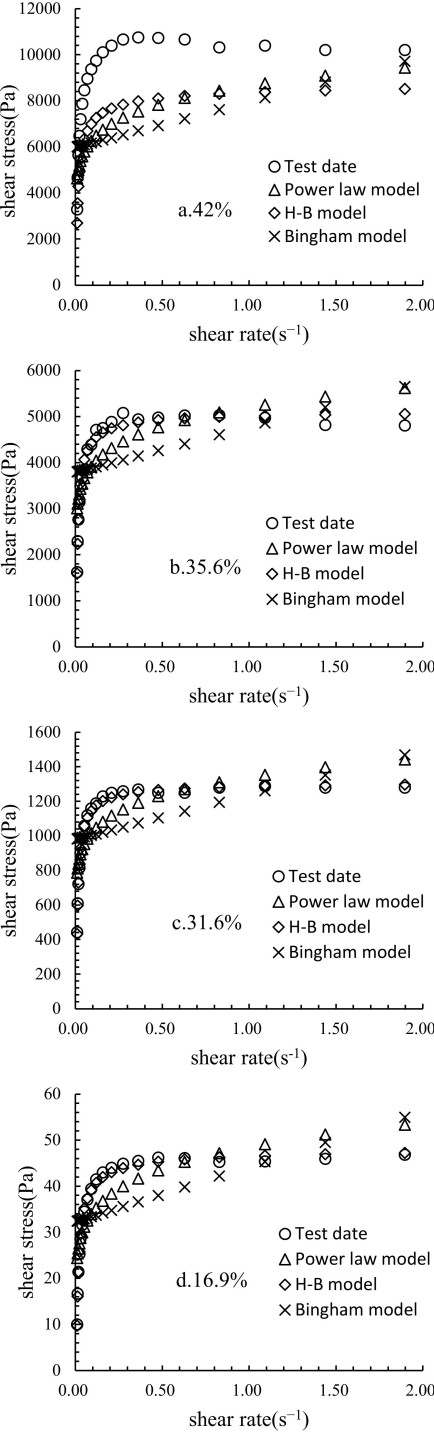

**Figure 3.** Experimental value and fitting value of the relationship between shear stress and shear rate of Chengdu clay mud with different water content.

In order to explore the suitable rheological model of Chengdu clay mud, the rheological experimental data were numerically fitted by the power law model, Bingham model and H–B model. Due to space limitations, some fitting results are drawn in Figure 3a–d, respectively (Figure 3a solid volume concentration 42%; Figure 3b solid volume concentration 35.6%; Figure 3c solid volume concentration 31.6%; Figure 3d solid volume concentration 16.9%). The rheological parameters and correlation coefficients obtained by fitting the rheological experimental results of slurries at different water contents by using different models are shown in Table 2.

**Table 2.** Rheological characteristic parameters and correlation coefficients obtained by fitting rheological data of slurries with different solid volume concentrations (corresponding to the shear rate in the range of $0.01 \sim 2.0 \text{ s}^{-1}$) for the Chengdu clay slurry.

| Solid Volume Concentration $C_s$ (%) | Yield Stress (Pa) | | | Viscosity (Pa.s) | | | Flow Index | | | Correlation Coefficient | | |
|---|---|---|---|---|---|---|---|---|---|---|---|---|
| | $\tau_{HB}$ | $\tau_B$ | $\tau$ | $\eta_{HB}$ | $\eta_B$ | $\eta$ | $n$ | | | $R$ | | |
| | H–B Model | Bingham Model | Power Law Model | H–B Model | Bingham Model | Power Law Model | H–B Model | Bingham Model | Power Law Model | H–B Model | Bingham Model | Power Law Model |
| 42.6 | 8889.7 | 5994.4 | - | −519.4 | 1967.1 | 8664.1 | −0.538 | 1 | 0.1350 | 0.992 | 0.596 | 0.889 |
| 39.4 | 5633.8 | 4269.1 | - | −96.6 | 1120.2 | 5790.1 | −0.791 | 1 | 0.1120 | 0.995 | 0.565 | 0.847 |
| 35.6 | 3157.2 | 3790.0 | - | −128.0 | 980.7 | 5202.7 | −0.721 | 1 | 0.1192 | 0.992 | 0.510 | 0.835 |
| 34.0 | 2099.2 | 1440.1 | - | −97.0 | 496.8 | 2085.6 | −0.587 | 1 | 0.1332 | 0.998 | 0.642 | 0.896 |
| 31.6 | 1315.8 | 981.7 | - | −28.5 | 257.8 | 1341.0 | −0.746 | 1 | 0.1160 | 1.00 | 0.5422 | 0.845 |
| 26.8 | 318.9 | 232.4 | - | −11.0 | 65.3 | 320.0 | −0.632 | 1 | 0.1166 | 0.997 | 0.605 | 0.886 |
| 22.0 | 181.3 | 114.5 | - | −11.0 | 47.0 | 177.0 | −0.562 | 1 | 0.1586 | 0.996 | 0.604 | 0.878 |
| 16.9 | 48.5 | 32.3 | - | −2.0 | 12.0 | 48.5 | −0.645 | 1 | 0.1496 | 0.998 | 0.572 | 0.859 |
| 10.9 | 12.8 | 8.0 | - | −0.5 | 3.0 | 12.9 | −0.690 | 1 | 0.1610 | 0.996 | 0.551 | 0.839 |
| 8.5 | 6.1 | 4.1 | - | −0.2 | 1.5 | 6.2 | −0.722 | 1 | 0.1510 | 0.996 | 0.549 | 0.835 |

In application of rheology, researchers try to use rheological models to describe the rheological curves within all shear rate ranges (Bingham model is the most popular one and it is widely used). Based on the experimental data and model fitting results of solid volume concentration mud in Figure 3, it is found that Bingham model (Equation (2)) can only briefly express the relationship between mud shear stress and shear rate (shear stress increases with the increase in shear rate), which cannot describe the relationship between stress and shear rate at a low shear rate. When the magnitude of shear rate exceeds $1.0 \text{ s}^{-1}$, the fitting value of shear stress is greater than the experimental data, and the fitting correlation coefficient r for each solid volume concentration mud is only between 0.5–0.6 (Table 2). The results of research [20,33] demonstrated that the Bingham model is more suitable to obtain the yield stress inferred from the linear part of the flow curve and plastic viscosity, rather than describe the rheological behavior of the whole stress–strain curve. As [30] emphasized, Bingham model overestimates the shear stress at lower strain rate.

The power law model (Equation (1)) can better reflect the non-linear growth process of mud shear stress with the increase in shear rate, but the description of the relationship between stress and shear rate at low shear rate is incomplete. When the magnitude of shear rate exceeds $1.0 \text{ s}^{-1}$, similar to Bingham model, the fitting value of shear stress is greater than the experimental data, and its correlation coefficient r is between 0.83–0.9 (in Table 2).

Compared with other models, the Herschel and Bulkley model (Equation (3)) has great advantages. For Chengdu clay mud with various solid volume contents, the fitting value of shear stress basically coincides with the experimental data, and it can well reflect the whole process of mud shear stress and shear rate (shear thinning). For each solid volume concentration, the fitting correlation coefficient r of Chengdu clay slurry is approximately one or even equal to one (the solid volume concentration is 31.6% Chengdu clay slurry). Therefore, based on the mathematical expression of mud shear stress and shear rate, it can be found that Herschel and Bulkley model (Equation (3)) is the best rheological model of Chengdu clay mud. This result is consistent with the research conclusions obtained by [8,13,30,34].

According to the fitting results (Table 2), the magnitude of flow coefficient $\eta_{HB}$ and flow index $n$ in the Herschel and Bulkley model are negative, which is quite different from

the common flow coefficient and flow index which are positive. Therefore, it is thought that this model is not applicable to simulate the rheological behavior of mud by the relevant literature, and other models with general fitting are still applied to describe the rheological process. According to the results of other fitting results above, it can be found that there is a great difference between the fitting value of shear stress and the actual shear stress. Due to the limitations of the author's knowledge, a more reasonable explanation cannot be afforded. However, for an accurate description of this process, an accurate mathematical model has more practical significance, it is vital for the complex interaction process between coarse particles and mud.

### 4.3. Effect of Solid Volume Concentration on the Yield Stress

According to the above analysis of fitting results, the Herschel–Bulkley model is adopted as the rheological model of Chengdu clay mud to analyze the influence of solid volume concentration on rheological parameters and yield strength. The following conclusions are drawn in Table 2: as solid volume concentration (from 42.6% to 8.5%) decreases, the yield strength decreases from 8889.7 pa to 6.1 pa. [8,35] found that parameters in Herschel–Bulkley model are strongly influenced by mud solid volume concentration. The research results of Chengdu clay mud are compared with those of volcanic ash mud [20]. The yield stress increases exponentially with the increase in particle concentration. In the semi-logarithmic coordinate systems, the increase coefficient of Chengdu clay mud is 0.2177, which is lower than that of the Nocera debris flow (0.4217) and the Montefiorino Irpino debris flow (0.6479). It may be due to the influence of different material composition in the Chengdu clay, the Nocera debris flow and the Montefiorino Irpino debris flow (Chengdu clay has more fine particles and less coarse particles), it means the influence of solid volume concentration on the yield stress of the Chengdu clay slurry. Nocera debris flow is less than of Montefiorino Irpino debris flow.

### 5. Conclusions

Based on the rheological experiments of Chengdu clay slurry with different solid volume concentrations, the influence of solid volume concentration on shear stress is analyzed in the paper. Different rheological models (the power law model, the Bingham model and the H–B model) are applied to fit the rheological process of Chengdu clay slurry with different solid volume concentrations, and the following conclusions are drawn:

1.  Chengdu clay mud represents typical shear thinning non-Newtonian behavior, and the influence of mud solid concentration on flow curve is quite different. When the magnitude of solid volume concentration is not less than 34% and the magnitude of shear rate is less than $1.0 \, \text{s}^{-1}$, the shear stress increases rapidly with the increase in the shear rate, while when the shear rate is greater than $1.0 \, \text{s}^{-1}$, the shear stress decreases with the increase of the shear rate. When the magnitude of solid volume concentration is less than 31.6% and the magnitude of shear rate is less than $1.0 \, \text{s}^{-1}$, the shear stress increases with the increase of shear rate, while when the magnitude of shear rate is more than $1.0 \, \text{s}^{-1}$, the shear stress is less affected by shear rate.
2.  In the range of low shear rate (less than $1.0 \, \text{s}^{-1}$), the increasing range of shear stress (slope of semi logarithmic coordinate flow curve) increases with the increase in solid volume content.
3.  The Herschel and Bulkley model can well reflect the whole process of shear stress and shear rate of Chengdu clay mud, and the correlation coefficient r is approximate to 1. The Herschel and Bulkley model is the best mathematical model to fit the rheology of Chengdu clay mud.

**Author Contributions:** Conceptualization, writing—original draft preparation, X.J.; formal analysis, Y.L.; methodology, W.C. All authors have read and agreed to the published version of the manuscript.

**Funding:** We acknowledge the financial support for this work, provided by the National Natural Science Foundation of China (Grant No. 41672357), Interdisciplinary Sciences Project, Nanyang Institute of Technology, Key R & D.

**Institutional Review Board Statement:** Not applicable.

**Informed Consent Statement:** Not applicable.

**Conflicts of Interest:** The authors declare no conflict of interest.

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
