# Peer review of "Effect of Solid Volume Concentration on Rheological Properties of Chengdu Clay Slurry"

_processes, doi:10.3390/pr10020425_

Round 1

Reviewer 1 Report

The authors present an interesting work that is appropriate for the  Processes journal. The main problem with this article is the writing in English, which must be reviewed by a native speaker.
In any case, here are some minor corrections:
References are not in journal format
In the introduction, the objective of the research is not well defined or is vague
In the graphs, the units must be placed in parentheses.
Line 111, there should be an error “Ip is: , it b”
Line 130, change Tab.1 by Table 1
Line 134 write with words infinity, change RPM to rpm
Line 139 (and others in the document) change don't use =, redact the phrase
In general, throughout the document there are certain spaces that are either missing or excessive.
Line 190 do not use the > sign, redact the phrase. In general, this error is repeated throughout the document and must be taken care of.
Table 2 is not in format
5. Conclusion – Change to 5. Conclusions
Some parts of the “Author Contributions” are missing

Author Response

Dear Reviewer:

Thank you very much for your valuable comments on this article. This article adds a lot of color because of your suggestion, The author gives the following reply to your suggestion:

Point 1: The authors present an interesting work that is appropriate for the Processes journal. The main problem with this article is the writing in English, which must be reviewed by a native speaker.

In any case, here are some minor corrections:

Response 1: The authors have carefully read the full text, partially revised the language in the text, and invited professionals with high English language level to guide the revision.

Point 2: References are not in journal format

Response 2: The references have been revised according to the format requirements (see the text)

Point 3: In the introduction, the objective of the research is not well defined or is vague

Response 3: The introduction has been revised to further clarify the research objectives

Point 4: In the graphs, the units must be placed in parentheses.

Response 4: The units in the chart have been modified

Point 5: Line 111, there should be an error “Ip is: , it b”

Response 5: "Line 11" has been modified

Point 6: Line 130, change Tab.1 by Table 1

Response 6: Modified

Point 7: Line 134 write with words infinity, change RPM to rpm

Response 7: Modified

Point 8: Line 139 (and others in the document) change don't use =, redact the phrase

Response 8: Modified

Point 9: In general, throughout the document there are certain spaces that are either missing or excessive.

Response 9: Modified

Point 10: Line 190 do not use the > sign, redact the phrase. In general, this error is repeated throughout the document and must be taken care of.

Response 10: Modified

Point 11: Table 2 is not in format

Response 11: P Modified

Point 12: 5. Conclusion – Change to 5. Conclusions

Response 12: Modified

Point 13: Some parts of the “Author Contributions” are missing

Response 13: Modified

Reviewer 2 Report

Please describe the shape of the solid particles (sphericity / angularity) and the proportion of each in the experiment (solid volume concentration)

Author Response

Dear Reviewer:

Thank you very much for your valuable comments on this article. This article adds a lot of color because of your suggestion, The author gives the following reply to your suggestion:

Point 1: Please describe the shape of the solid particles (sphericity / angularity) and the proportion of each in the experiment (solid volume concentration)

Response 1: The purpose of this paper is to explore the rheological characteristics and applicable rheological model of Chengdu clay slurry. The samples are taken from Longquan District, Chengdu. Only the particle size of the samples has been analyzed, but the particle shape and their respective proportions have not been analyzed. According to your suggestion, the author carefully consulted the relevant literature and supplemented the relevant contents about the shape of clay particles in Chengdu.